# Molecular Cloning and Expression Responses of *Jarid2b* to High-Temperature Treatment in Nile Tilapia (*Oreochromis niloticus*)

**DOI:** 10.3390/genes13101719

**Published:** 2022-09-25

**Authors:** Min Zhou, Zhilei Yao, Min Zhao, Qingfeng Fang, Xiangshan Ji, Hongju Chen, Yan Zhao

**Affiliations:** Shandong Provincial Key Laboratory of Animal Biotechnology and Disease Control and Prevention, Shandong Agricultural University, Taian 271000, China

**Keywords:** Nile tilapia, high-temperature, *Jarid2b*, cloning, expression pattern

## Abstract

Nile tilapia is a GSD + TE (Genetic Sex Determination + Temperature Effect) fish, and high-temperature treatment during critical thermosensitive periods (TSP) can induce the sex reversal of Nile tilapia genetic females, and brain transcriptomes have revealed the upregulation of *Jarid2* (Jumonji and AT-rich domain containing 2) expression after 36 °C high-temperature treatment for 12 days during TSP. It was shown that JARID2 forms a complex with polycomb repressive complex 2 (PRC2) that catalyzed H3K27me3, which was strongly associated with transcriptional repression. In this study, *Jarid2b* was cloned and characterized in Nile tilapia, which was highly conserved among the analyzed fish species. The expression of *Jarid2b* was upregulated in the gonad of 21 dpf XX genetic females after 12-day high-temperature treatment and reached a similar level to that of males. Similar responses to high-temperature treatment also appeared in the brain, heart, liver, muscle, eye, and skin tissues. Interestingly, *Jarid2b* expression was only in response to high-temperature treatment, and not to 17α-methyltestosterone (MT) or letrozole treatments; although, these treatments can also induce the sex reversal of genetic Nile tilapia females. Further studies revealed that *Jarid2b* responded rapidly at the 8th hour after high-temperature treatment. Considering that JARID2 can recruit PRC2 and establish H3K27me3, we speculated that it might be an upstream gene participating in the regulation of Nile tilapia GSD + TE through regulating the H3K27 methylation level at the locus of many sex differentiation-related genes.

## 1. Introduction

Sex determination refers to the process by which sexually reproducing organisms determine and initiate the differentiation from an early undifferentiated gonad toward the testis or ovary [1,2]. The sex of fish is remarkably plastic [3]. According to the role of genetic factors and environmental temperature in sex determination, the sex determination mechanisms in fish can be broadly divided into the following three major categories: genotypic sex determination (GSD), temperature-dependent sex determination (TSD), and genotypic sex determination with temperature effects (GSD + TE) [4]. For many fishes with GSD + TE, such as Nile tilapia (*Oreochromis niloticus*), the sex is jointly regulated by genetic factors and environmental temperature, and the environmental temperature can affect sex determination and differentiation only in a certain time interval, which is called the thermosensitive period (TSP). During TSP, the artificial treatments of high or low temperatures result in sex-ratio changes [5,6,7]. For example, artificially high-temperature treatment at 36 °C from approximately 9 days post-fertilization (dpf) and lasting from 10 to 30 days could induce the sex reversal of XX genotypic females to XX pseudomales in Nile tilapia [8,9,10], which masked the genetic XX/XY sex-determination system in Nile tilapia [11,12,13].

*Jarid2* (Jumonji and AT-rich domain containing 2) is characterized by a conserved Jumonji C (JmjC) domain and is one of the JmjC domain protein family members [14]. JARID2 can form a complex together with polycomb repressive complex 2 (PRC2) that is vital to recruit polycomb helper proteins to the target genes and catalyze the methylation of histone H3 at lysine 27 to yield H3K27me3, which is strongly associated with transcriptional repression [15,16]. JARID2 was also proven to regulate histone methyltransferase complex activity and it could coordinate the control of PRC2 occupancy and enzymatic activity at target genes in early embryos and embryonic stem cells (ESCs) [14,17,18,19,20]. Through the regulation of epigenetic states of target genes, the JARID2-PRC2 complex was thought essential to regulate the development and differentiation of embryonic cells and tumor cytogenesis in human, mice, and bovine [21,22,23,24,25]. Additionally, JARID2 could also regulate the pluripotency and differentiation of ESCs by the strict expression regulatory of *Nanog* and *β-catenin* [26]. Interestingly, in our previous studies, brain transcriptome analysis revealed that *Jarid2* showed a male-biased expression pattern, and high-temperature treatment of XX genetic females during TSP, resulted in a male rate of 83.47%, accompanied by the significant upregulation of the *Jarid2* expression level and reaching the similar level as the males during this period [4]. Recent studies have shown that the sexually dimorphic expression of genes in the brain may play an essential role in response to gonadal differentiation, which might consequentially or causatively respond to fish gonadal sex [27,28]. Therefore, we presume that *Jarid2* may play an important role in gonad differentiation in Nile tilapia or other fish species. Additionally, the 11th intron of *Jarid2* was observed to preferably be retained in nearly all transcripts from ZZf (ZZ female) tissues in alligator, turtle, or bearded dragon. It was speculated that intron retention may be involved in regulating the expression level of functional JARID2 protein and might be crucial in the TSD of some reptiles [29,30,31]. It is unknown whether intron retention of *Jarid2* plays an important role in fish with TSD or GSD + TE. So far, the changes in the *Jarid2* expression level during TSP after high-temperature treatment have not been determined in the various species exhibiting TSD or GSD + TE, and whether *Jarid2* is an important cue for sex reversal after high-temperature treatment in Nile tilapia is still unknown. Therefore, it is meaningful to detect and analyze the specific expression pattern of *Jarid2* during high-temperature-induced female sex reversal in Nile tilapia.

The purpose of this study was to clone the *Jarid2b* cDNA sequence, analyze protein sequence conservation, construct the phylogenetic tree, and examine the tissue expression pattern, to investigate the effect of high-temperature treatment during TSP on its mRNA and protein expression levels in the gonads of tilapia larvae using qRT-PCR, Western blot, and IHC and to further compare the effects of high-temperature treatment with the treatment of the exogenous sex steroid hormone (17α-methyltestosterone, MT) and aromatase inhibitor (Letrozole) on *Jarid2b* expression. 

## 2. Materials and Methods

### 2.1. Ethics

The study was approved by the Shandong Agricultural University Animal Care and Use Committee with approval number SDAUA-2015-017. All surgeries were performed under tricaine methane sulfonate solution (MS222) (Sigma, Beijing, China) anesthesia, and all efforts were made to minimize the suffering of the Nile tilapia.

### 2.2. Fish Reproduction

A total of 24 two-year-old Nile tilapia (270.26 ± 13.53 g) was cultured in our laboratory, including twelve females (XX♀), six sex-reversed pseudomales (XX♂), and six males (XY♂), whose parents were collected from the Guangxi Fisheries Institute (Nanning, Guangxi, China), and were used in this study. Pseudomales (XX♂) were obtained by feeding the diets containing low concentrations of MT during the sex differentiation stage to induce the sex reversal of Nile tilapia genetic females (XX♀). The genotypic sex of females, pseudomales, and males was identified using screened sex-linked RAPD-SCAR marker [32,33]. The phenotypic sex of pseudomales was defined by the identification of external reproductive organs. When Nile tilapia was sexually mature, all-XX embryos were produced by culturing one female plus one pseudomale in the 360 L tank under natural photoperiod and the water temperature ranged from 26 to 29 °C. The embryos at about 5–7 days post-fertilization (dpf) in the female mouth were taken out and cultivated independently in the 30 L aquaria as a family with a water temperature of 28 ± 0.5 °C. In the same way, three normal families were produced by culturing one XX female and one XY male in three tanks, respectively, and the larvae in normal families contained XX and XY individuals with a ratio of about 1:1.

### 2.3. Larvae Culturing and Sampling

The embryonic condition was observed daily and the high-temperature treatment that promoted male development was started when yolk absorption was completed at about 9 dpf. The high-temperature-induced masculinization of the Nile tilapia was performed as previously described [7,34]. The all-XX embryos from each family were randomly and equally divided into two groups rearing in 20 L tanks, respectively, and 100 larvae were incubated in each tank. The two groups were the control female group (XX) and the high-temperature-treated female group (XX + HT), and larvae from three families were regarded as three biological replicates. Larvae in the XX group were always reared at 28 °C and the water temperature of the XX + HT group gradually elevated to 36 °C in 4 h and was maintained at 36 °C for 12 days (from 9 to 21 dpf). The larvae from three normal families were reared in three tanks at 28 °C from 9 to 21 dpf, respectively, and 21 dpf genetic XY larvae were identified using the sex-linked RAPD-SCAR marker [33].

The brain tissues of 10 larvae from each family in the XX and XX + HT groups were harvested separately and mixed at 4, 6, 8, 10, 12, 15, 18, and 21 h after high-temperature treatment. The samples were placed in liquid nitrogen and later transferred to a −80 °C refrigerator for subsequent experiments. Similarly, the heart, brain, muscle, liver, eye, gill, skin, and gonad were collected from 10 individuals at 21 dpf in the XX, XX + HT, and XY groups. In the same way, the eye, heart, liver, muscle, spleen, skin, brain, ovary, and blood samples from three 180 dpf adult XX fish and the brain and testis samples from three 180 dpf adult XY fish culturing at 26–29 °C were sampled, respectively.

### 2.4. MT and AI Treatments

Another three all-XX families and three normal families were developed in the same way as described before. The larvae in each all-XX family were divided into four groups and each group contained 120 larvae. Nile tilapia XX larvae were cultured at 28 °C from 9 to 21 dpf and were fed three times per day with a powdered diet sprayed with 95% ethanol only (named as XX, the first group) or a diet sprayed with 95% ethanol containing MT (Solarbio, Beijing, China) at a concentration of 20 μg/g diet (named as XX + MT, the second group), which is the lowest MT concentration achieving a 100% male ratio [35]. The 9 dpf fry were cultured in 28 °C water containing 40.5 μg/L letrozole (MedChem Express, USA) (named as XX + AI, the third group) for 12 days, which can obtain a high male ratio [36]. Additionally, all-XX Nile tilapia larvae in another group were cultured at 36 °C from 9 to 21 dpf and fed a diet sprayed with 95% ethanol only (named XX + HT, the 4th group). The 120 larvae from each normal family were cultured at 28 °C from 9 to 21 dpf and fed the diet sprayed with 95% ethanol only, the genetic XY larvae were identified at 21 dpf (named as XY). All fish were reared under the same conditions, except for food and water temperature. At 21 dpf, the gonads of 50 fish from each group were sampled.

### 2.5. Molecular Cloning and Bioinformatics Analysis of Tilapia Jarid2b

Total RNA was extracted from the tissue samples mentioned above using the RNAsimple Total RNA Kit (TIANGEN, Beijing, China), respectively, according to the manufacturer’s instructions. RNA integrity, concentration, and purity were detected using the agarose gel electrophoresis and a Nanodrop 2000 spectrophotometer (Thermo Fisher Scientific, Wilmington, DE, USA), respectively. The cDNA was synthesized using Evo M-MLV RT Kit (Accurate Biotechnology, Changsha, China).

*Jarid2b*-specific primers (Table 1) for PCR were designed using Primer 6.0 based on the sequences obtained by transcriptome analysis (SRP159698) in our laboratory [4]. PCR was performed using 2× Accurate Taq Master Mix (Accurate Biotechnology, Changsha, China) and the amplification procedure was as follows: 94 °C (30 s), followed by 35 cycles of 98 °C (10 s), 55 °C (30 s), and 72 °C (5 min) and final extension at 72 °C (5 min). Then the PCR product was purified and subcloned into the pMD™ 18-T Vector (Takara, Japan). Finally, the recombinant plasmid was transformed into *E. coli* DH5α competent cells (Takara, Shiga, Japan) to obtain positive clones, followed by validation via DNA sequencing. 

The molecular weight, theoretical isoelectric point, and average hydrophilicity were calculated by SMS2 [37]. Predictive analysis of *Jarid2b* signal peptide cleavage sites was performed using SignalP 3.0 [38]. The domain architecture prediction of Nile tilapia *Jarid2b* was performed using SMART [39]. The amino acid sequence of Nile tilapia *Jarid2b* was aligned with other species using the DNAMAN programs (Lynnon Biosoft, Quebec, QC, Canada). Phylogenetic analysis based on the amino acid sequences of *Jarid2b* was conducted using MEGA software (version 7.0.14, Mega Limited, Auckland, New Zealand) by the neighbor-joining method with 1000 bootstraps.

### 2.6. qRT-PCR Analysis

*Jarid2b* and *β-actin* specific primers (Table 1) for qRT-PCR were designed and the cDNA samples obtained above were used as the templates. qRT-PCR assays were performed using SYBR^®^ Green Premix Pro Taq HS qPCR Kit in a total 20 μL reaction volume according to the manufacturer’s instructions (Accurate Biotechnology, Changsha, Hunan, China). qRT-PCR cycling conditions in Roche LightCycler^®^ 96 were followed as initial denaturation at 95 °C for 30 s, and then 40 cycles of denaturation at 95 °C for 5 s and extension for 30 s at 60 °C, followed by disassociation curve analysis to determine target specificity. The relative expression of the *Jarid2b* gene was calculated based on the delta-delta Ct method and normalized to the *β-actin* mRNA level [36,40]. PCR specificity was assessed by melting curve analysis.

### 2.7. Western Blot Analysis

Anti-*Jarid2b* polyclonal antibody was produced by Sangon Biotech Co., Ltd. (Shanghai, China), and commercial β-ACTIN (Sangon, Shanghai, China) and GAPDH (Sangon, Shanghai, China) antibodies were used in this study. The protein samples were extracted from the tissues mentioned above (each about 5 mg). Next, the protein concentration was detected using the BCA protein assay kit (Beyotime Biotechnology, Shanghai, China), and the samples reached the same concentrations by adding ddH_2_O before use. Then, the protein samples were run on a 6% SDS–PAGE gel and were transferred to PVDF membranes. Subsequently, the membranes were incubated with the rabbit anti-JARID2B, anti-β-ACTIN, or anti-GAPDH antibody diluted at a ratio of 1:1000 with 10 mM PBS (pH 7.4) overnight at 4 °C and then incubated with a horseradish peroxidase (HRP)-labeled goat anti-rabbit secondary antibody (Beyotime Biotechnology, Shanghai, China) of 1:1000 at room temperature for 3 h. Finally, the JARID2B protein band was visualized using BeyoECL Plus (Beyotime Biotechnology, Shanghai, China) on the protein toning system (Vilber Lourmat, Paris, French).

### 2.8. Immunohistochemistry Analysis

The ovary and testis of 180 dpf Nile tilapia females and males were fixed in Bouin’s solution to dehydrate for 24 h at room temperature, embedded in paraffin wax, and sectioned at 5 μm thickness for immunohistochemistry (IHC). Firstly, the sections were deparaffinized, hydrated, and blocked with 3% H_2_O_2_ at room temperature for 1.5 h. Secondly, the slides were incubated with the anti-JARID2B antibody of 1:1000 overnight at 4 °C and then incubated with the HRP-labeled secondary antibody mentioned above at room temperature for 1 h. Next, an enhanced HRP-DAB Chromogenic Kit was applied for enzymatic reactions (Beyotime Biotechnology, Shanghai, China). Finally, the slides were stained in hematoxylin and visualized on Zeiss confocal microscope.

### 2.9. Statistical Analysis

All data were expressed as the average ± SD (n = 3). One-way analysis of variance (ANOVA) and Tukey’s test were used to analyze the data using SPSS 21. Differences were considered significant when *p* < 0.05. Relative expression of *Jarid2b* gene was plotted using GraphPad Prism 8.0 software (version 8.0.2.263, GraphPad Software Inc., San Diego, CA, USA).

## 3. Results

### 3.1. Molecular Cloning and Bioinformatic Analysis of Jarid2b 

PCR using the specific primer pair was performed to obtain the complete open reading frame (ORF) of *Jarid2b*. The full-length *Jarid2b* ORF in Nile tilapia was 4239 bp and encoded a 1412-amino acid protein with a calculated molecular weight of 154.98 kDa and a theoretical isoelectric point of 10.14. The average hydrophilicity was −0.739 and the signal peptide was not found in *Jarid2b*. SMART online software analysis showed that three conserved and key structural features were found in Nile tilapia *Jarid2b*, including Jumonji N (JmjN) domain, BRIGHT and ARID domain, and JmjC domain (Figure 1).

Multiple sequence alignments based on the degrees of homology at the protein level showed that Nile tilapia JARID2B shared high homology with JARID2B in other fish species including *Oreochromis aureus* (XP_031598248.1), *Astatotilapia calliptera* (XP_026011421.1), *Maylandia zebra* (XP_024654161.1), *Pundamilia nyererei* (XP_005728530.1), *Simochromis diagramma* (XP_039877292.1), *Haplochromis burtoni* (XP_014187632.2), *Archocentrus centrarchus* (XP_030596394.1), *Melanotaenia boesemani* (XP_041867268.1), and *Amphiprion ocellaris* (XP_023136762.1), and the sequence identity was, respectively, 99.50%, 96.65%, 96.58%, 98.23%, 97.66%, 97.95%, 93.71%, 88.56%, and 89.94% (Figure 2). Thus, the JARID2B amino acid sequence in Nile tilapia was highly similar to that of *O.*
*a.* and *P. n.* Furthermore, the JmjN domain, BRIGHT and ARID domain, and JmjC domain are relatively conserved in these analyzed species (Figure 2).

The predicted Nile tilapia JmjN domain (690–731) consists of 42 amino acid residues with the same sequence as *O. a.* and *M. b*. Alanine (A) residue at position 718 in predicted Nile tilapia JmjN domain was substituted for Serine (S) or Tyrosine (Y) only in *A. centrarchus* and *A. o*. Valine (V) residue at position 724 in predicted Nile tilapia JmjN domain was replaced with A in *A. calliptera*, *M. z.*, *P. n*., *S. d.,* and *H. b.* and both V and A occurred with the half probability among these ten species analyzed. The predicted JmjC domain (1032–1196) comprised 165 amino acid residues, which were the same as that in *O. a*. Aspartic acid (D) residue at position 1109 tended to be displaced with Asparagine (N) in half the analyzed species. In *M. b.* and *A. o.*, the amino acid residues at position 1041 changed from N to A or Threonine (T), and at position 1100 changed from I to V.

The phylogenetic tree constructed using MEGA 7.0 revealed that JARID2B and JARID2A could be clustered in respective clades. The JARID2B in Nile tilapia gathered into a cluster with the one in *O. a.* and later with *H. b.* and *S. d.* (Figure 3), showing that the JARID2B in Nile tilapia had the highest homology with the one in *O. a*.

### 3.2. Tissue Expression Distribution of Jarid2b

The tissue expression distribution of *Jarid2b* was analyzed using cDNAs synthesizing from the total RNA isolated from 11 tissues of Nile tilapia at 180 dpf. *β-actin* expression in Nile tilapia was examined as an internal reference. The results showed that *Jarid2b* was constitutively expressed in all the examined tissues (eye, spleen, heart, liver, blood, XX brain, XY brain, muscle, skin, ovary, and testis) (Figure 4A). The highest expression level of *Jaird2b* was observed in Nile tilapia blood, followed by brain and testis, while the lowest was in the ovary. Moderate expression levels were detected in the eye, spleen, heart, liver, muscle, and skin. Western blot analysis of 21 dpf Nile tilapia gonads showed that the anti-JARID2B polyclonal antibody only stained the 155 kD protein band in the XX, XX + HT, and XY samples as expected, which verified the specificity of the JARID2B antibody (Figure 5B). Furthermore, the titer of the antibody reached 1:512,000. The specific JARID2B antibody was used for Western blot analysis and the results showed that the JARID2B protein expression profile was similar to that obtained from the qRT-PCR (Figure 4B).

### 3.3. High-Temperature, but Not MT/letrozole Treatment, Upregulates Nile Tilapia Gonadal Jarid2b mRNA and Protein Expression

To investigate the expression pattern of *Jarid2b* after various treatments, the *Jarid2b* levels in the Nile tilapia gonad at 21 dpf after treatment for 12 days were examined in this study. According to qRT-PCR, the expression of the *Jarid2b* gene in the XX + HT and XY groups was significantly higher than that in the XX group at 21 dpf (Figure 5A). There was no significant difference between the XX + HT and XY group, showing the important role of high-temperature treatment in affecting the expression of *Jarid2b*. In addition, Western blot analysis showed a similar JARID2B protein expression profile to those obtained from the qRT-PCR (Figure 5B), and the high-temperature treatment also up-regulated the JARID2B protein expression level. Consistent with the Western blot data, the IHC results showed that weak positive signals were observed in somatic cells in the XX gonad, whereas strong positive signals were observed in Sertoli cells in the XX + HT and XY gonads (Figure 5C). Collectively, high-temperature treatment during TSP resulted in a significant upregulation of Nile tilapia *Jaird2b* mRNA and protein expression.

Sex steroid hormones and their inhibitors can affect the expression of a large number of genes in many species by activating related nuclear receptors. To explore whether MT and letrozole can affect *Jarid2b* expression, qRT-PCR assays were performed using the sampled gonads after various treatments. Letrozole treatment (XX+ AI) of XX genotypic females of Nile tilapia larvae at 9 dpf for 12 days (21 dpf) did not affect the mRNA level of *Jarid2b* in the gonads compared to the XX control group (Figure 6). Similarly, MT treatment (XX+ MT) did not also affect the expression of *Jarid2b*. Collectively, contrary to the effects of high-temperature treatment on the expression of *Jarid2b* in the gonads (XX + HT), MT and letrozole treatments do not affect the expression of *Jarid2b* in the gonads. Therefore, the expression of *Jarid2b* in the Nile tilapia gonads was specifically affected by high-temperature treatment during TSP.

### 3.4. High-Temperature Treatment Also Upregulates Jarid2b Expression in Other Tissues in Nile Tilapia

High-temperature treatments during TSP upregulated the mRNA and protein expression of *Jaird2b* in the gonad of Nile tilapia. However, whether the expression of *Jaird2b* in various tissues was affected by high-temperature treatment remains to be elucidated. The results showed that the expression of gene-encoding *Jarid2b* in the heart, brain, liver, muscle, eye, and skin in the XX + HT group was significantly increased compared with that in the XX group (Figure 7). The highest upregulation of *Jarid2b* was observed in the eye of the XX + HT group with about 52-fold increases, while the lowest upregulation was in the skin with about 2.5-fold increases. However, no significant change in *Jarid2b* expression was observed in the gill after high-temperature treatment (Figure 7). In conclusion, the expression of gene-encoding *Jarid2b* in various tissues except for gill was significantly upregulated after high-temperature treatment.

### 3.5. Jarid2b Expression Responds Early to High-Temperature Treatment 

Because high-temperature treatment affected the expression of *Jarid2b* in most tissues in Nile tilapia, the brain tissue was selected to investigate whether the *Jarid2b* expression responds early to high-temperature treatment. Temporal changes in brain *Jarid2b* transcript levels within 21 h after high-temperature treatment were examined and the result showed that the expression levels of *Jarid2b* had no significant differences at 4 and 6 h after high-temperature treatment in the XX + HT group compared with the XX group, and then turned to extremely significant upregulation from 8 to 21 h (Figure 8). Taken together, this result indicated that high-temperature treatment during TSP could affect the transcript level of *Jarid2b* as early as 8 h.

## 4. Discussion

### 4.1. The Identification and Tissue Distribution of Jarid2b

In this study, we successfully isolated and sequenced *Jarid2b* in the Nile tilapia. The result of bioinformatic analysis and multiple sequence alignments indicated that *Jarid2b* belonged to the JmjC gene family and had a highly conserved JmjC domain located at residues 1032–1196. The JmjC domain was first defined based on the amino-acid similarities in the JARID2 (Jumonji), JARID1C (Smcx), and JARID1A (RBP2) proteins [41,42,43]. JmjC-domain-containing proteins were classed as seven evolutionarily conserved groups including the JHDM1, PHM2/PHF8, JARID, JHDM3/JMJD2, UTX/UTY, JHDM2, and JmjC domain only [21]. Nowadays, it is considered that JmjC-domain-containing proteins might be involved in demethylation within histones [44,45]. However, JARID2, one subgroup of the JARID group, was predicted to have no histone demethylase activity because it did not share the conserved residues that were essential for histone demethylase activity compared to other JmjC-domain-containing proteins [21,46]. Moreover, the function of the JmjC domain also seemed to be different within the two JARID subgroups, as the amino acids required for enzymatic function are intact in most members of the JARID1 subgroup but completely lacking in the JARID2 subgroup [46]. Further study showed that *Jarid2* was certified to constitute a subunit of PRC2 and related to the catalytical activity of histone methylation [14]. In this study, Phylogenetic analysis showed that the *Jarid2b*-deduced amino acid sequence was conserved among the analyzed fish species and Nile tilapia *Jarid2b* was most similar with that of *O. a.*. JARID2B was relatively conservative between different species, which implied that its potential histone methylation activity may be conserved across multiple species. JmjN is one of the conserved Jmj domains and the function of the JmjN domain remains largely undetermined. Research has demonstrated that JmjN and JmjC interact physically to form a structural unit that ensures the stability activity of *Gis1* [47]. Moreover, the JmjN in *Jhd2* is also important for its protein stability [48]. So far, there were no reports related to the function difference between *Jarid2a* and *Jarid2b*. Nile tilapia *Jarid2a* gene has also not been reported.

Herein, we investigated *Jarid2b* gene expression patterns in adult Nile tilapia tissues and revealed differential expression levels in various tissues, which is consistent with the obtained expression pattern based on transcriptome data in Nile tilapia [49]. There is a variety of evidence that *Jarid2* is widely expressed and has different biological functions in other species. Studies have shown that *miR-130a*, an *Etv2* downstream target, was defined an important role in the mediation of vascular patterning and angiogenesis. Mechanistically, miR-130a directly regulated *Jarid2* expression by binding to its 3′-UTR region, and its expression was increased in zebrafish *miR-130a* morphants. Further study showed that over-expression of *Jarid2* in HUVEC cells led to defective tube formation indicating its inhibitory role in angiogenesis. These findings demonstrated a critical role for *Etv2*-*miR-130a*-*Jarid2* in vascular patterning [50]. Additionally, the *Jarid2* expression profile during embryonic development and in adult tissues in mangrove rivulus fish suggested that it might be important in development, gametogenesis, and neurogenesis, which may be related to the epigenetic regulation role of *Jarid2* [51]. Moreover, *Jarid2* expression at high levels was detected in the heart, brain, thymus, and skeletal muscle in adult mouse and in the brain and heart in adult human, indicating that it might be involved in multiple organ development [21,52,53,54].

In this study, we found that *Jarid2b* was expressed in the gonads of both sexes during the TSP and the expression pattern exhibited sexual dimorphism, which suggested that it may be involved in the early development and differentiation of Nile tilapia gonads. Remarkably, the sexually dimorphic patterns of *Jarid2b* transcripts appeared to be maintained to adulthood as demonstrated by the tissue distribution analysis in this study, suggesting that it may be important for gonadal differentiation and maintaining the adult sexual phenotype in Nile tilapia.

### 4.2. High-Temperature, but Not MT/letrozole Treatment, Affects Jarid2b Expression 

In this study, the effect of high-temperature treatment on *Jarid2* expression was first studied in detail in GSD + TE fish. The results showed that *Jarid2b* expression was notably high in the gonad of XX tilapia after high-temperature treatment compared to the XX control at 21 dpf, which suggests that *Jarid2b* may play an important role in the high-temperature induced sex reversal of Nile tilapia females into pseudomales. The use of specific androgen or aromatase inhibitors can cause the sex reversal of genetic females into phenotypic males in various fish species exhibiting GSD + TE or GSD, and affect the expression of many hormone receptor or steroid synthase-related genes [35,36,55,56,57]. For instance, genetic all-female rainbow trout were treated with the androgen 11beta-hydroxyandrostenedione, which resulted in 100% males at a dosage of 1 mg/kg in food. Steroid enzyme P450scc was clearly up-regulated, and 3betaHSD and P450aro were down-regulated during the treatment [55]. In zebrafish, an all-male population was observed after exposure to 9.7 ng/L synthetic androgen trenbolone and above from 20 to 59 days post-hatch (dph) [58]. Another study showed that batches of tilapia fry treated with aromatase inhibitor during the first 30 days following yolk-sac resorption (7–37 dph) and the percentage of males remained approximately constant (92.5–96.0%) from 200 to 500 mg/kg [59]. Quantitative analysis showed that a certain concentration of MT or letrozole treatment resulted in a similar sex reversal rate of Nile tilapia XX genetic females as the high-temperature treatment [35,36]. However, the results in this study showed that the same doses of MT or letrozole treatment as used by Teng et al. [36] and Wang et al. [35] did not affect *Jarid2b* expression, which is different from the result of high-temperature treatment. So far, the effect of MT or letrozole treatment on *Jarid2* expression has not been reported. High-temperature, but not MT/letrozole treatment, affects *Jarid2b* expression, which suggests that the molecular mechanism and the pathways of sex reversal in Nile tilapia females induced by androgen or aromatase inhibitor treatment perhaps are partially different from that of high-temperature treatment. We speculated that high-temperature and MT/letrozole treatment may mutually act on some downstream genes [36], but the upstream genes affected by high-temperature and MT/letrozole treatment may be completely different.

After high-temperature treatment during TSP, the level of *Jarid2b* transcripts in most tissues of Nile tilapia were significantly upregulated. For example, the average *Jarid2b* mRNA level in juvenile tilapia blood in the XX + HT group was 52 times higher than that in the XX group. High-temperature treatment also affected the expression of *jarid2b* in multiple tissues except for gills, such as the heart and brain, suggesting that *Jarid2b* may be a constitutively expressed gene and play an important role in coping with temperature treatment in most tissues. We speculated that the molecular mechanism of *Jarid2* expression regulated by high-temperature treatment is basically the same in multiple tissues. Similarly, it was shown that the expression of *Hsp70* and *Hsp90* was up- or down-regulated in gill, liver, and muscle when Kaluga (*Huso dauricus*) was treated with different temperatures or salinities [60]. qRT-PCR showed that the upregulated response of turbot (*Scophthalmus maximus*) PRLR at multiple time points (1 h, 6 h, 12 h, 24 h, 3 m, and 9 m) was similar in gill, kidney, and intestinal tissues after low salt (5, 10, or 30 ppt) treatment [61]. 

Previous studies showed that the treatment with the glioma inhibitory drug temozolomide (TMZ) resulted in *Jarid2* downregulation and CCND1 upregulation within glioma tissues of different grades, and further studies showed that JARID2 negatively regulates CCND1 expression by increasing the H3K27me3 level on the CCND1 promoter in leukemia cell [62,63]. In addition, *Jarid2* expression was increased in bladder cancer tissues and cells, and upregulation of *Jarid2* increased the H3K27me3 level at the PTEN promoter, thus enhancing the progression of bladder cancer through regulating PTEN/AKT signaling [64]. Given that *Jarid2* could generally regulate H3K27me3 status, we speculate that *Jarid2b* may affect the methylation level of genes in the female differentiation pathway and thereby suppress their expression during the sex differentiation in Nile tilapia of GSD + TE. Recently, Zhong et al. [65] found a high H3K27me3 level could transcriptionally repress the expression of RUNX1 (the runt-related transcription factor 1), a transcription factor influencing granulosa cells’ growth and ovulation, whereas RUNX1 acts as an activator of steroidogenesis-related genes *Cyp19a1*, promoting the production of estrogen in porcine. Similarly, Lee et al. [66] provided in vivo evidence that the level of H3K27me3 is involved in the rapid changes in *Cyp19a1* expression by altering the chromatin structure of the promoters. Furthermore, CBX2, a subunit of the Polycomb Repressive Complex 1 (PRC1), which can mainly regulate the level of H3K27me3, can directly bind the ovary-promoting gene *Lef1*, resulting in the bivalent and repressed status in Sertoli cells of the XY fetal gonad. These results suggested that stabilization of the testis fate requires H3K27me3-mediated repression of ovary-determining genes, which would otherwise block testis development in mice [67].

### 4.3. Jarid2b Expression Was Affected Early by High-Temperature Treatment 

*Jarid2b* expression in the brain responded sharply in the early period of high-temperature treatment during TSP and its expression was significantly upregulated from 8 h. In *Arabidopsis*, the cold-related COR (cold-regulated) gene was mediated by CRT (C-repeat)/DRE (dehydration-responsive element). CBF1, a transcriptional activator, was found to bind to the CRT/DRE and its overexpression could induce COR gene expression and increase freezing tolerance [68,69,70]. The transcript levels for CBF increased within 15 min after transferring plants to a low temperature, followed by the accumulation of COR gene transcripts at about 2 h, which indicated that CBF gene induction is an early event in the low temperature-stimulated signaling cascade [71,72]. Similarly, the rapid expression responses of barley clock genes to temperature were examined and the clock genes such as CCA1 and PRR73 responded rapidly to the changes in temperature within 6 h [73,74]. Pufferfish (*Takifugu rubripes*) HSP70 showed a rapid response to temperature treatment (from 24 to 28 °C or 24 to 20 °C) at the 3rd h in the gill, muscle, and liver, speculating that HSP70 might act as the main gene to regulate fish adaptive capacity with changed temperature [75]. Thus, these genes, which rapidly respond to temperature changes, may be the upstream gene responding to high-temperature treatment and the connection point between temperature treatment and downstream genes affected by temperature treatment. Therefore, we speculate that *Jarid2b* may be an upstream gene responding to high-temperature treatment and play an important role in the regulation of Nile tilapia GSD + TE.

## 5. Conclusions

In this study, *Jarid2b* was characterized in Nile tilapia, and *Jarid2b* was observed to be commonly expressed in multiple tissues in adult tilapia and exhibited a male-biased expression pattern. We have shown for the first time that high-temperature treatment, but not MT or AI treatment, upregulated *Jarid2b* levels in the gonads of juvenile Nile tilapia at 21 dpf, and this upregulation was consistent across multiple tissues. *Jarid2b* expression was found to rapidly respond to high-temperature treatment. Our results suggest that *Jarid2b* may play an important role in the regulation of Nile tilapia GSD + TE.

## Figures and Tables

**Figure 1 genes-13-01719-f001:**
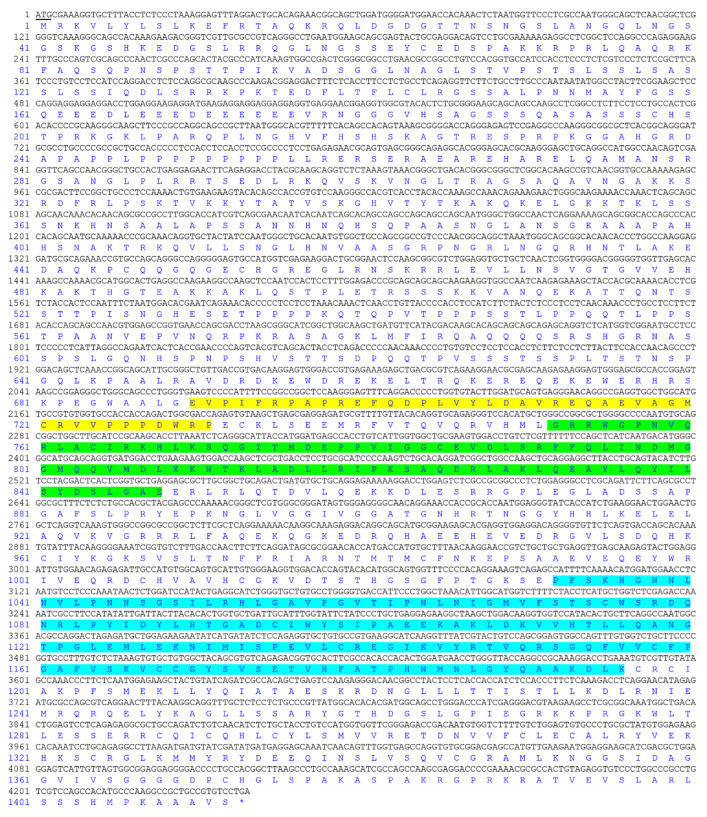
Complete coding sequence and deduced protein sequence of *Jarid2b* in Nile tilapia (GenBank accession number: ON571717). Positions of nucleotides and amino acids are labeled on the left. The predicted JmjN domain, BRIGHT and ARID domain, and JmjC domain are shaded in yellow, green, and blue-grey, respectively. Underline represents the initiation codon. The stop codon is marked by an asterisk (*).

**Figure 2 genes-13-01719-f002:**
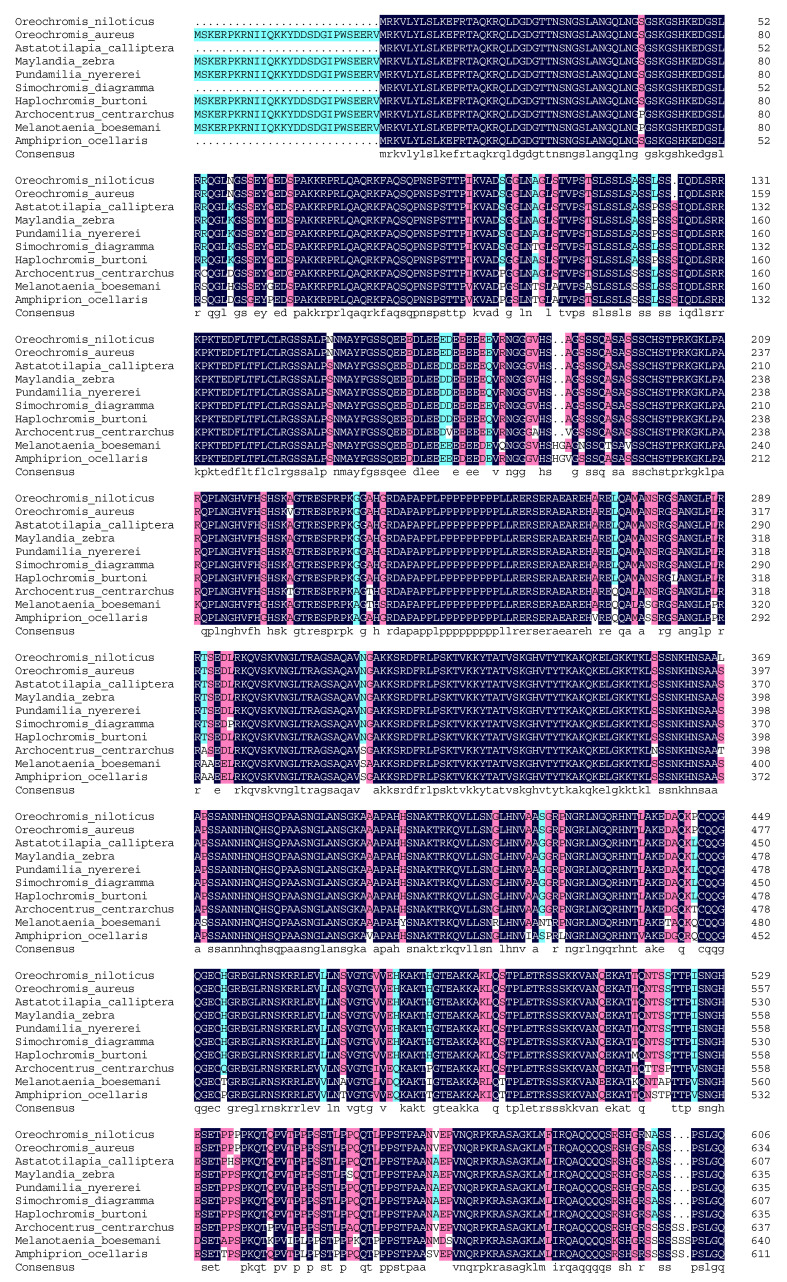
Multiple alignments of JARID2B amino acid sequence in Nile tilapia and other species. The identical amino acid residues are shaded in dark blue-grey. Pink shade indicates highly conserved amino acid residues and the amino acid residues only in one or two species differed from those of other species. The predicted JmjN domain, BRIGHT and ARID domain, and JmjC domain are marked with red overlines.

**Figure 3 genes-13-01719-f003:**
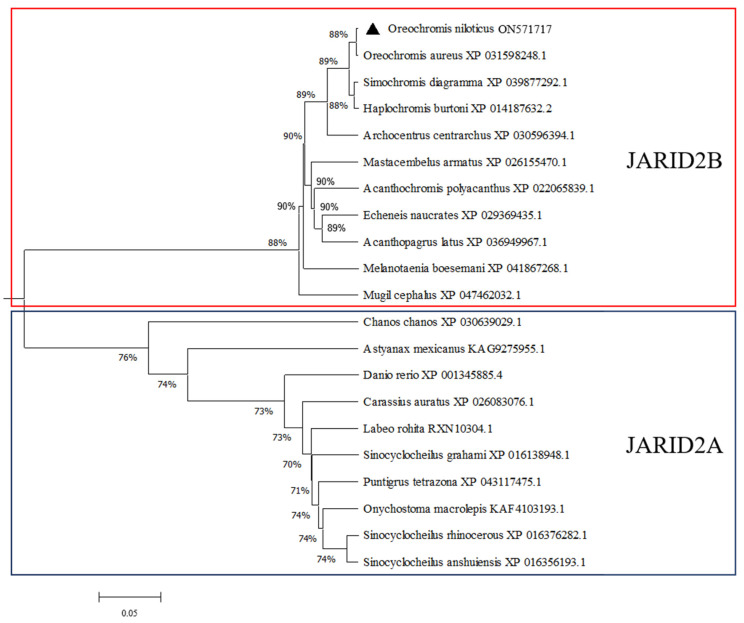
Phylogenetic analysis of JARID2B and JARID2A from different species. Protein sequence IDs are indicated behind the species name. Numbers at the branch of the phylogenetic tree stand for bootstrap. Nile tilapia *Jarid2b* is marked with a black triangle (▲).

**Figure 4 genes-13-01719-f004:**
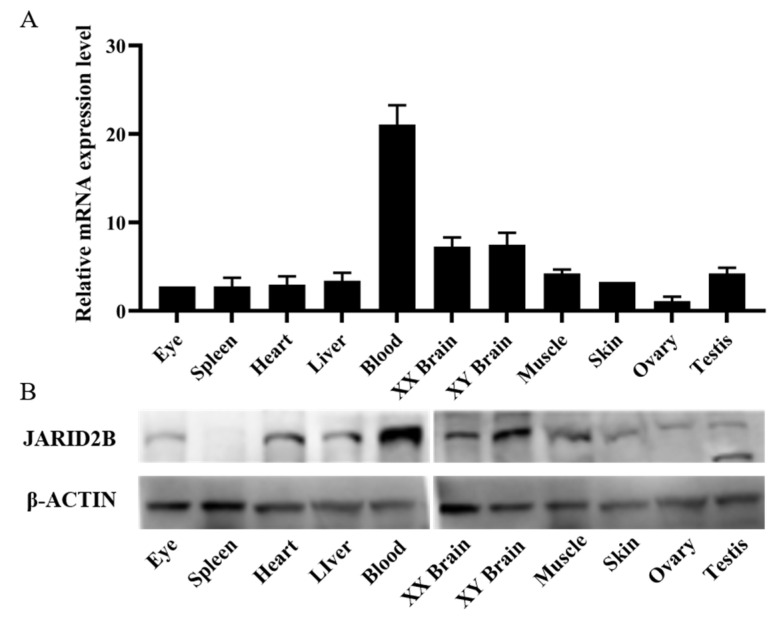
mRNA and protein expression profiles of *Jaird2b* in different tissues of Nile tilapia at 180 dpf: (**A**): Relative expression of *Jarid2b* gene as determined by qRT-PCR. All values are the mean ± SD; n = 3. The expression level of *Jarid2b* in the ovary was set as 1. (**B**): Protein expression of *Jarid2b* as determined by Western blot.

**Figure 5 genes-13-01719-f005:**
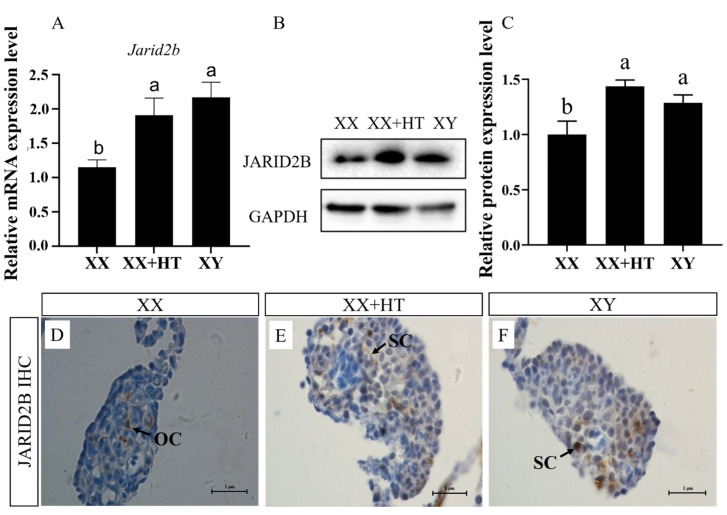
Effects of the high-temperature treatment on the abundance of *Jarid2b* in Nile tilapia gonads at 21 dpf: (**A**): Relative expression of *Jarid2b* gene at 21 dpf as determined by qRT-PCR. The expression level of *Jarid2b* in the XX group was set as 1. The different lower-case letters indicate significant differences between treatments (*p* < 0.05, ANOVA). XX: control female group; XX + HT: high-temperature-treated female group; XY: control male group. (**B**): Protein expression of *Jarid2b* at 21 dpf as determined by Western blot. (**C**): Band intensity quantification in Figure 5B was performed in Image J software (version 1.8.0.172, National Institute of Mental Health, Bethesda, Maryland, USA), and GAPDH was used as a loading control. The different lower-case letters indicate significant differences between treatments (*p* < 0.05, ANOVA). (**D**–**F**): IHC results of fish from the XX, XX + HT, and XY groups at 21 dpf. OC: somatic cells; SC: Sertoli cells. The positive signal is brown.

**Figure 6 genes-13-01719-f006:**
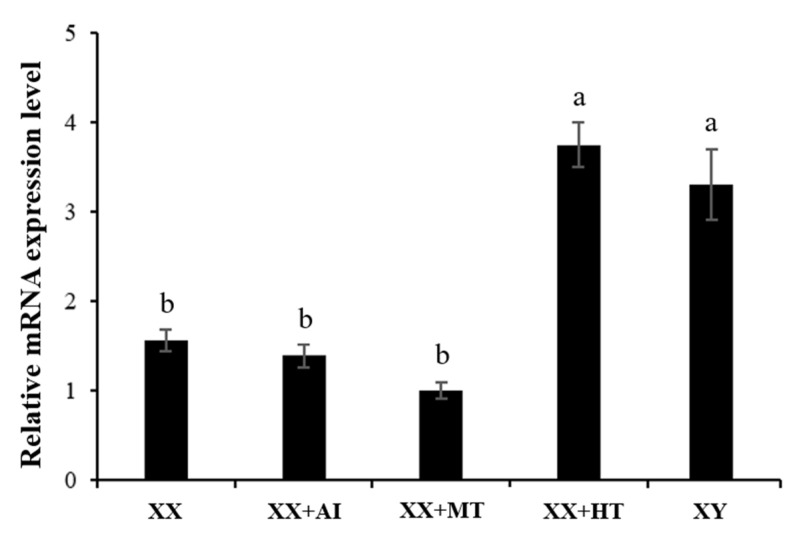
Gonadal *Jarid2b* expression responds to 17-Methytestosterone (MT) and letrozole (AI) treatment at 21 dpf. XX + MT: genetically XX Nile tilapia with 17-Methyltestosterone treatment under 28 °C temperature water; XX + AI: genetically XX Nile tilapia with letrozole (AI) treatment under 28 °C temperature water; XX: control female group; XX + HT: high-temperature-treated female group; XY: control male group. The expression level in the gonad of the XX group was defined as 1, following normalization to the *β-actin* expression. The different letters indicate significant differences among treatments (*p* < 0.05).

**Figure 7 genes-13-01719-f007:**
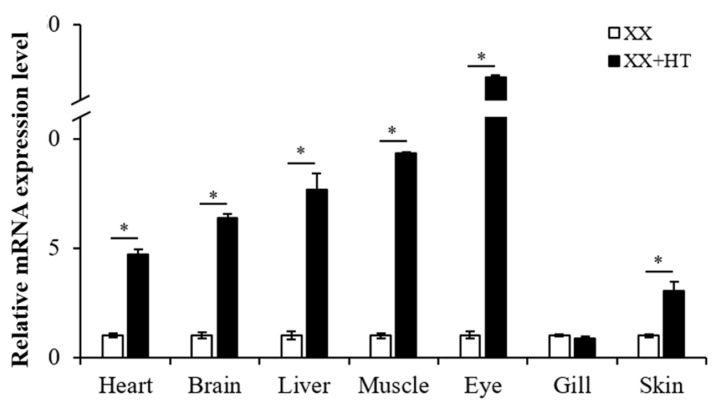
Expression of *Jarid2b* gene in various tissues as determined by qRT-PCR. XX: control female group; XX + HT: high-temperature treated female group. The expression level of *Jarid2b* in each tissue in the XX group was set as 1. * Statistically significant difference (*p* < 0.05).

**Figure 8 genes-13-01719-f008:**
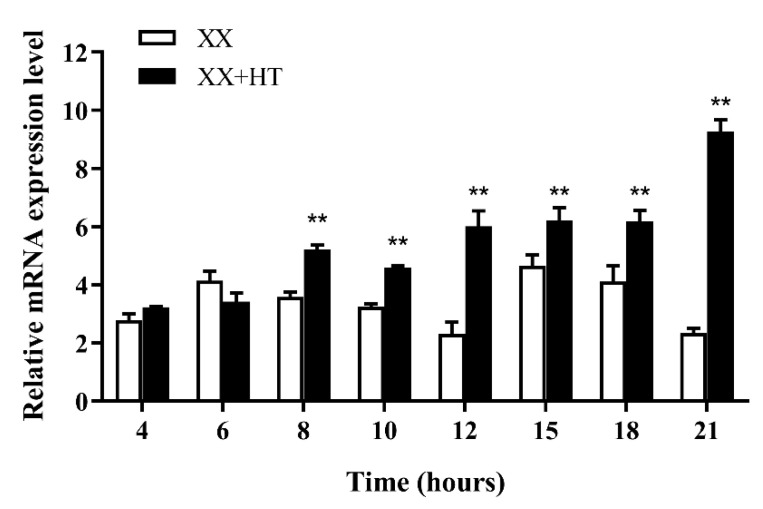
Chronological expression of *Jarid2b* in XX larvae brains within 21 h after high-temperature treatment. ** Statistically significantly difference (*p* < 0.01).

**Table 1 genes-13-01719-t001:** Sequence of primers.

Primer Pairs	Primer Sequence (from 5′ to 3′)	Amplicon Length/bp	Purpose
*Jarid2b*-F1	ATGCGAAAGGTGCTTTACCTCTCC	4239	Cloning
*Jarid2b*-R1	TCAGGACACGGCAGCGGCCTTGGG		
*Jarid2b*-F2	GAAGGCATCAAGGTTTATCG	221	qRT-PCR
*Jarid2b*-R2	GCGATCTGATACAGTAGCTT		
*β-actin*-F	TGACCTCACAGACTACCTCATG	224	qRT-PCR
*β-actin*-R	GGCAACGGAACCTCTCATTG		

## Data Availability

Not applicable.

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
