# Peer review of "Molecular Cloning and Expression Responses of Jarid2b to High-Temperature Treatment in Nile Tilapia (Oreochromis niloticus)"

_genes, 2022, doi:10.3390/genes13101719_

Round 1

Reviewer 1 Report

Comments:

Manuscript genes-1900356 is an interesting work providing information about Molecular cloning and expression responses of Jarid2b to high-temperature treatment in Nile tilapia. However, the current form of ms is not suitable yet for publication as some critical issues found in the text. The ms should be improved by addressing those issues before being accepted for publication. Several parts of text need to be carefully rewritten for English use, expression, etc. The issues and comments are listed as follows:

Title

Ln 3 – should be “Nile tilapia (Oreochromis niloticus)”

Abstract 

Ln 19 – The font size of word “letrozole” should be same as other word. 

Introduction

Lns 30-34 – Rewrite the sentence!

Lns 34-39 – Rephrase the sentences!

Lns 68-73 – See the previous comment regarding the font size of the word. Check it in the entire text and make it consistent!

Lns 76-82 – It is better to write these sentences without numbering.

Materials and Methods

Ln 89 – Did the authors measure the size of the fish, such as the total length (average and standard deviation)?

Ln 118 – Did the authors euthanize the fish before dissection? Mention it in the ms!

Ln 120 – Did the authors add the reagent for stabilize and protect RNA with immediate RNase inactivation (RNA stabilization and storage) to the samples?

Ln 154 – The authors used a subclone pMD™ 18-T vector which is 2692bp in size, and the authors only used one primer for cloning purposes. However, the full-length Jarid2b ORF in Nile tilapia was 4239 (see ln 209 in the results section). Please clarify this!

Ln 157 – a) Mention the GenBank accession number and the references where the primers designed from!

      b) Target size for each pair of primers should be included in Table 1

Ln 168 –Did the authors conduct the melting curve analysis for the exclusion of primer combinations forming primer/dimers and specificity confirmation of newly designed primers? State it in ms!

Lns 174-175 – Include the references in order to support this statement

Results 

Ln 209 – See the previous comment ln 154.

Ln 216 – It is strongly recommended that the gene sequence data obtained in this study should submitted to the public databases such NCBI and state its reference number into the text.

Lns 275-277 – Rewrite the sentences and state the significance based on the p-value!

Ln 282 – “mRNA” should be “expression of jarid2b gene” or “expression of gene encoding jarid2b”.  Apply this for entire text!

Lns 294-298 –  a)        Mention the complete legend, such as the significance based on the p-value etc.

b) State which organs are in figures C, D, and E.

c) Write the caption for figures D and E.

Discussion

Lns 349-354 –This part is more like a results section. Modify the sentence.

Reviewer 2 Report

General points

The article entitled “Molecular cloning and expression responses of Jarid2b to high-temperature treatment in Nile tilapia” characterized Jarid2b (Jumonji and AT-rich domain containing 2) in Nile tilapia. The expression of Jarid2b was investigated in various tissues during high-temperature treatment, which showed considerable changes but not during treatment with 17α-methyltestosterone (MT) or letrozole.

The manuscript is worthy of investigation. The manuscript is interesting, well-written, and discussed. Herein, I recommend a minor revision.

I almost do not have concerns or comments, except for one point.

Specific points:

1-     Line 175: why did the authors normalize the expression results using β-actin only? It is preferable and more accurate to use more than one reference gene. In addition, other reference genes may be preferable; please refer to Yang et al., 2013; DOI: 10.1016/j.gene.2013.06.013.

Round 2

Reviewer 1 Report

Comments:

The manuscript (genes-1900356-peer-review-v2) has been well-improved according to the

previous comments and I would recommend it for publication.
